# High-dimension Prototype is a Better Incremental Object Detection Learner

**Yanjie Wang[1], Liqun Chen[1], Tianming Zhao[1], Tao Zhang[1], Guodong Wang[2],**
**Luxin Yan[1], Sheng Zhong[1], Jiahuan Zhou[3,\*], Xu Zou[1,\*]**
[1]National Key Laboratory of Multispectral Information Intelligent Processing Technology,
Huazhong University of Science and Technology, Wuhan 430074, China
[2]College of Computer Science and Technology, Qingdao University, Qingdao 266071, China
[3]Wangxuan Institute of Computer Technology, Peking University, Beijing 100871, China
`{aiawyj, chenliqun, tming, zhangtao, yanluxin, zhongsheng}@hust.edu.cn`
`doctorwgd@gmail.com, jiahuanzhou@pku.edu.cn, zx@zoux.me`

## Abstract

Incremental object detection (IOD), surpassing simple classification, requires the simultaneous overcoming of catastrophic forgetting in both recognition and localization tasks, primarily due to the significantly higher feature space complexity. Integrating Knowledge Distillation (KD) would mitigate the occurrence of catastrophic forgetting. However, the challenge of knowledge shift caused by invisible previous task data hampers existing KD-based methods, leading to limited improvements in IOD performance. This paper aims to alleviate knowledge shift by enhancing the accuracy and granularity in describing complex high-dimensional feature spaces. To this end, we put forth a novel higher-dimension-prototype learning approach for KD-based IOD, enabling a more flexible, accurate, and fine-grained representation of feature distributions without the need to retain any previous task data. Existing prototype learning methods calculate feature centroids or statistical Gaussian distributions as prototypes, disregarding actual irregular distribution information or leading to inter-class feature overlap, which is not directly applicable to the more difficult task of IOD with complex feature space. To address the above issue, we propose a Gaussian Mixture Distribution-based Prototype (GMDP), which explicitly models the distribution relationships of different classes by directly measuring the likelihood of embedding from new and old models into class distribution prototypes in a higher dimension manner. Specifically, GMDP dynamically adapts the component weights and corresponding means/variances of class distribution prototypes to represent both intra-class and inter-class variability more accurately. Progressing into a new task, GMDP constrains the distance between the distribution of new and previous task classes, minimizing overlap with existing classes and thus striking a balance between stability and adaptability. GMDP can be readily integrated into existing IOD methods to enhance performance further. Extensive experiments on the PASCAL VOC and MS-COCO show that our method consistently exceeds four baselines by a large margin and significantly outperforms other SOTA results under various settings.

## 1 Introduction

In recent years, deep learning methods [3; 61] have witnessed remarkable advancements in various visual tasks, particularly in object detection [30; 58]. However, these methods typically learn defined labeled classes from static datasets, limiting their applicability in dynamic real-world scenarios. Incremental Object Detection (IOD) methods have emerged as an exciting and challenging task to address this issue. In contrast to human incremental learning, deep learning often suffers from the problem of catastrophic forgetting [45], where new information interferes with previous knowledge. This phenomenon poses a significant challenge for IOD, complicating the retention and integration of new data without disrupting previously learned information.

---

\*Corresponding author.

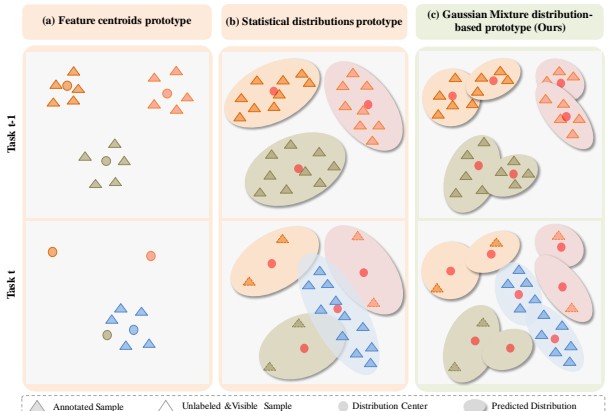

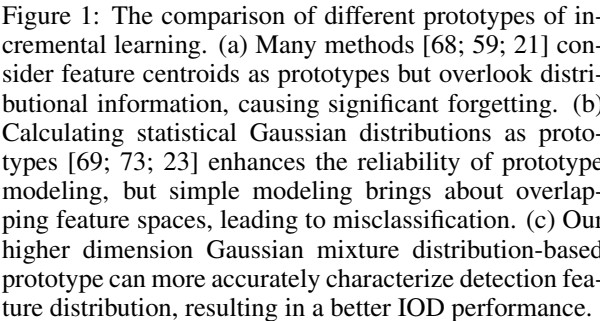

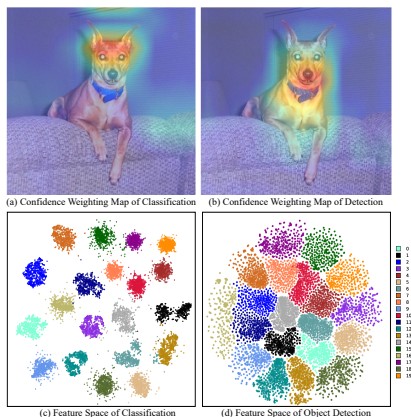

Figure 1: The comparison of different prototypes of incremental learning. (a) Many methods [68; 59; 21] consider feature centroids as prototypes but overlook distributional information, causing significant forgetting. (b) Calculating statistical Gaussian distributions as prototypes [69; 73; 23] enhances the reliability of prototype modeling, but simple modeling brings about overlapping feature spaces, leading to misclassification. (c) Our higher dimension Gaussian mixture distribution-based prototype can more accurately characterize detection feature distribution, resulting in a better IOD performance.

Figure 2: Confidence weighting maps and t-SNE [60] visualization of embedding for classification and object detection. (top) In contrast to classification focusing on the local salient region, detection also emphasizes global shape and boundary features. (bottom) Classification tasks reveal clear feature separation, while detection tasks present distinct, crowded distributions, rendering detection features more complex.

Existing IOD methods can be roughly divided into two paradigms: Rehearsal-based and Knowledge Distillation (KD)-based IOD. Rehearsal-based IOD [4; 34; 35] aims to mitigate catastrophic forgetting by reutilizing previously training samples. The efficacy of these methods hinges upon the replay strategy of the samples. In contrast, KD-based IOD [48; 7; 72], the most commonly employed paradigm, leverages features and responses from a teacher model's previous class to guide a student model in acquiring new knowledge. Nonetheless, distilling new data could introduce knowledge bias due to the invisible previous task data, leading to limited improvements in IOD performance.

Recently, several works have explored Prototype Learning [49; 73; 41] to address catastrophic forgetting, which characterizes classes by learning the distribution of features within classes without keeping any previous task feature representations, thus compensating for the deficiency of KD. Existing incremental learning methods [19; 68; 59; 21] primarily rely on feature centroids as prototypes (Figure 1 (a)). However, this approach overlooks distributional information, resulting in significant forgetting due to insufficient modeling of previous data. As illustrated in Figure 1 (b), some methods [69; 73; 23] calculate statistical Gaussian distributions as prototypes to enhance the reliability of prototype modeling by considering class distributions, which to some extent improves detection performance. Due to limitations in modeling capabilities, the prototype based on a statistical Gaussian distribution fails to accurately estimate complex detection feature distributions, leading to confusion between current and prior knowledge since estimated distributions may overlap. From the perspective of confidence weighting maps, the features for classification in Figure 2 (a) are predominantly concentrated in local salient regions, whereas detection features (Figure 2 (b)) also attend to global boundaries or shape characteristics [6]. For t-SNE visualizations of classification(Figure 2 (c)), features from different classes often exhibit clear delineation, while those from the same classes demonstrate a Gaussian-like clustered distribution. Conversely, for detection tasks (Figure 2 (d)), features from different classes tend to be relatively crowded and exhibit distinct distribution patterns. Therefore, the above methods are tailored for incremental classification and are not directly applicable to the more challenging task of IOD, which involves a more complex feature space.

In this paper, we propose a novel higher-dimension-prototype learning approach for KD-based IOD to alleviate the knowledge shift. This approach enables a more flexible, accurate, and fine-grained representation of feature distributions without retaining any previous task data. To address the above issue of existing prototype learning methods, we propose a Gaussian Mixture Distribution-based

Prototype (GMDP), which explicitly models the distribution relationships between different classes by directly measuring the likelihood of embedding from new and old models into class distribution prototypes in a more complex manner to fit high-dimensional feature spaces of object detection, as shown in Figure 1 (c). Specifically, the GMDP dynamically adjusts the component weights and corresponding means and variances of class distribution prototypes to represent intra-class and inter-class variability accurately. Progressing into a new task, GMDP would constrain the distance between the distribution of new task classes and previous classes, minimize overlap with existing class features, and strike a balance between stability and adaptability. In addition, implementing GMDP in detection tasks involves unique challenges, particularly in managing the complexities of high-dimensional features during the learning process. To address this issue, we leverage a Length Scaling Progressive Learning (LSPL) method, which gradually learns complex, high-dimensional features by first focusing on the discriminative characteristics of new classes. To enhance the plasticity of class prototypes, we present a Dynamic Adaptive Prototype Optimization (DAPO) strategy for IOD. This strategy aims to enhance the adaptability and effectiveness of GMDP modeling by improving the separation of class features through inter-class mean dispersion, intra-class component cohesion, and prototype variance minimization. Simultaneously, our approach can improve performance further with Rehearsal-based methods. In summary, our contributions are as follows:

- To alleviate the knowledge shift, we propose a novel prototype learning approach for KD-based IOD, which enables a more flexible, accurate, and fine-grained representation of feature distributions without keeping any previous task data.

- A novel GMDP is introduced to explicitly model the distribution relationships between different classes suitable for more complex feature spaces. A Length Scaling Progressive Learning method is proposed to address the learning difficulties of GMDP. To enhance the plasticity of class prototypes, we present a DAPO strategy for IOD during the incremental process, which could enhance the adaptability and effectiveness of GMDP modeling.

- GMDP can be readily integrated into existing IOD methods to enhance performance further. Methods with our idea on the PASCAL VOC and MS COCO consistently outperform 4 baselines significantly and achieve state-of-the-art results in different settings.

## 2 RELATED WORK

### 2.1 INCREMENTAL LEARNING

Incremental learning [39; 9; 44] aims to continuously update a model by introducing different subsets of the label space without retraining the model on old data. Previous methods could be roughly categorized into rehearsal-based, parameter-isolation, and regularization-based fashion. Rehearsal-based fashion methods mitigate catastrophic forgetting induced by new classes by periodically revisiting and rehearsing previously learned samples [4; 28; 49], or generating new samples [26; 56; 64]. However, these methods suffer from challenges related to privacy protection and high storage costs. Parameter-isolation methods [37; 43; 42; 67] segregate parameters for different tasks to ensure that learning new tasks does not interfere with previously learned tasks. Regularization-based methods [39; 10; 11; 22] impose constraints on the features and parameters to balance learning between new and previous classes, thereby maintaining stability and generalization during the model updating process. In this work, we focus on regularization methods based on Knowledge Distillation.

### 2.2 CONTINUAL LEARNING WITH PROTOTYPES

Recently, various prototype-based class incremental learning (CIL) methods have been proposed to continually learn new classes without preserving any historical exemplars [55; 68; 73]. Some methods [1; 62; 74] treat the prototype of each class as a learnable embedding vector. Mean Feature Prototypes calculate the mean feature of all samples from the same class to form the prototype [21; 55; 73]. However, using a single feature center can lack informative distribution. To address this, the latest methods [16] calculate both the mean feature vector and its variance from all samples to better represent class distribution. These approaches, however, can be susceptible to outliers and class overlap, which may undermine stability due to misclassification risks [70; 71]. In this paper, we introduce the GMDP to constrain the distance between the distribution of new and previous classes, and minimize overlap with existing class features.

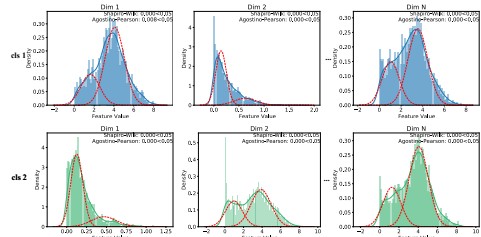 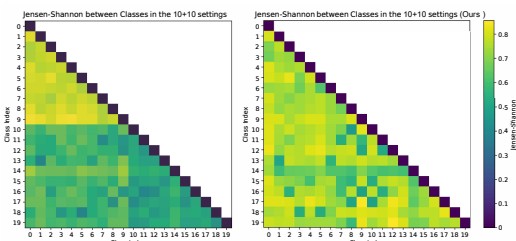

Figure 3: Histogram visualizations of the statistical distribution of detection features.

Figure 4: The confusion matrix of detection feature divergence across different classes.

## 2.3 INCREMENTAL OBJECT DETECTION

Differing from general object detection [52; 14; 51; 15; 50] with static training data, Incremental Object Detection (IOD) [57; 24; 65; 5; 13] aims to dynamically adapt to new tasks introduced over time, while mitigating the risk of catastrophic forgetting, which could be roughly categorized into the Rehearsal-based fashion and knowledge distillation-based fashion. Rehearsal-based fashion [36; 38] involves storing a balanced set of examples, such as images [36] or sample instances [38] and periodically revisiting previously learned samples at each incremental step to finetune the model. For knowledge distillation-based fashion, after Shmelkov *et al.* [57] introduced the first knowledge distillation-based incremental detector ILOD, most researchers [18; 8; 47; 27] have designed various knowledge distillation-based IOD methods on many object detection architectures.

Despite the KD-based IOD being the most commonly employed paradigm, distilling new data could introduce knowledge bias due to the invisible previous task data, leading to limited IOD performance. Conversely, prototype-based incremental learning methods encourage the model to learn discriminative class decision boundaries of new and old models. However, inaccurate prototype distribution modeling may result in overlapping feature distributions among classes in complex high-dimensional feature spaces. Given the complementary advantages of KD and prototype methods, we focus on a KD approach based on class distribution prototypes, balancing detector stability and adaptability by explicitly modeling complex detection features without keeping previous task data.

## 3 PROPOSED METHOD

### 3.1 MOTIVATION

In contrast to existing works [49; 73; 41] that only adopt recognition features for prototype computation, our method leverages both recognition and location-based detection features generated from the linear layer of RoI. Thus, we aim to capture the multi-faceted class characteristics as shown in Figure 2 (b), enabling a more comprehensive and nuanced representation of discriminative features. To illustrate the distribution of IOD detection features across diverse dimensions, we exemplify the features from class 1 and class 2 across various dimensions as shown in Figure 3. These features are extracted from the linear layer of RoI trained on the PASCAL VOC. Each subplot indicates the distribution of detection features across distinct dimensions. Employing Shapiro-Wilk [54] and Agostino-Pearson [46] tests, we evaluate the normality of each dimension feature, revealing deviations from Gaussian distribution. Consequently, IOD detection features are inherently in a more complex feature space, diverging from the simplicity of recognition features, posing challenges for single Gaussian distribution modeling [69; 73; 23]. Moreover, variations are observed between the dimensions of different classes. That is, each dimension of the features can be considered independently distributed, enabling more complex modeling. This encourages us to further validate the feasibility of applying more precise prototype representations of IOD. We obtained more suitable and accurate probability density estimates by fitting each dimensional feature adopting a Gaussian Mixture Distribution (Red dashed line) compared to a single Gaussian Distribution. Therefore, we introduce Gaussian Mixture Distribution-based Prototype (GMDP) as a more flexible and refined approach to represent IOD feature distribution by introducing distinct components for each class.

The effectiveness of GMDP plays a crucial role in representing class prototypes that overcome catastrophic forgetting and enhance model plasticity. To validate this, we follow the definition of distribution distance proposed by Liu et al. [2; 33] to distinguish classes and adopt the *Jensen-Shannon (JS) divergence* metric to quantify the distance between class distributions. The distance

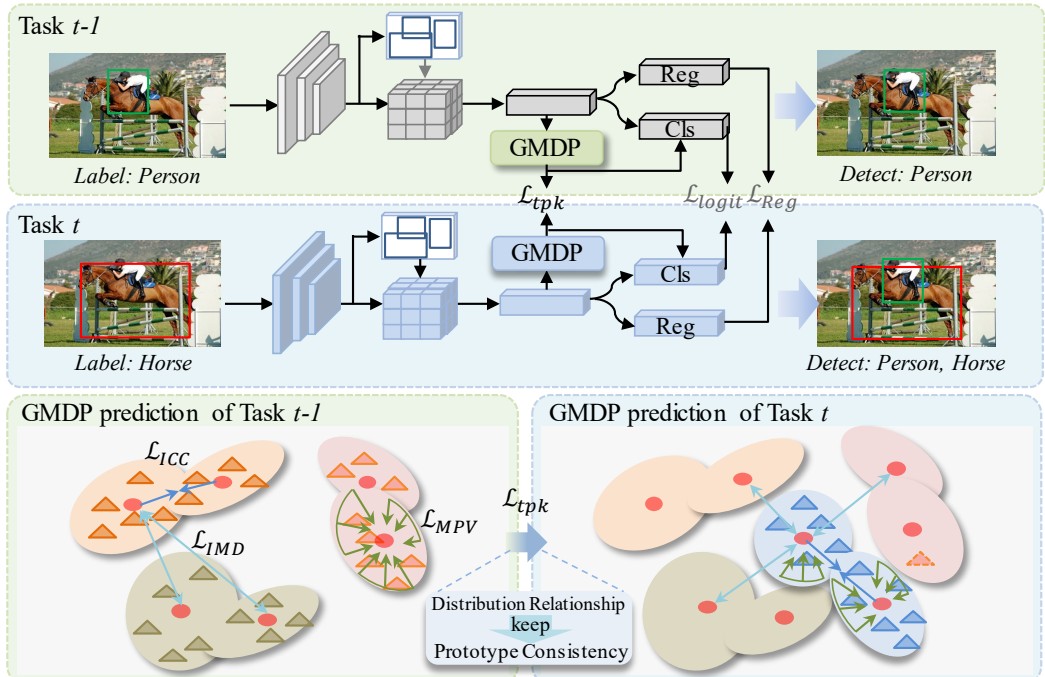

Figure 5: The overall framework. The Gaussian Mixture Distribution-based Prototype (GMDP) introduces a flexible, accurate feature characterization in a more complex manner to fit high-dimensional feature spaces. GMDP would transfer old prototype knowledge by constraining the distribution relationships of the prototype to maintain consistency, thereby alleviating catastrophic forgetting. To enhance the plasticity of prototypes, we further present a dynamic adaptive prototype optimization strategy ($\mathcal{L}_{IMD}$, $\mathcal{L}_{ICC}$ and $\mathcal{L}_{MPV}$) to improve the modeling adaptability of GMDP.

between the distribution of two classes can be computed as follows:

$$d_{JS} = \frac{1}{2}D(P||\mu) + \frac{1}{2}D(Q||\mu) \tag{1}$$

where $d_{JS}$ is the Jensen-Shannon divergence, $D$ refers to the KL divergence, $P$ and $Q$ are the Two distribution-based class prototypes, and $\mu = (P + Q)/2$ is the midpoint distribution. Figure 4 illustrates the distance between class prototypes guided by the statistical Gaussian distributions-based prototype (SGDP) (left) and our GMDP (right) in the 10-10 task. 10-10 task means that the first stage trains annotations containing 10 classes and then trains images containing the remaining 10 classes in the second stage on PASCAL VOC. A smaller divergence value indicates less distinguishability between two prototypes, potentially resulting in poorer detection performance. Notably, class prototypes guided by SGDP (Figure 4 left) exhibit smaller disparities with the 10 classes in the second task compared to other prototypes. Conversely, classes in both stages with our GMDP display better discriminability. Thus, our GMDP can more accurately describe the class distributions in complex feature space compared to SGDP, promoting the modeling of inter-class and intra-class differences.

## 3.2 GAUSSIAN MIXTURE DISTRIBUTION-BASED PROTOTYPE

To alleviate the knowledge shift issue, we propose a novel prototype learning approach for KD-based IOD as Figure 5, which can more flexibly, and accurately represent feature distributions without keeping any previous task data. The key idea is, based on the above analyses, to learn a set of Gaussian mixture prototypes that capture the essential features according to their distribution characteristics and adopt these prototypes to transfer knowledge from an old model to the new one.

Specifically, we introduce a Gaussian Mixture Distribution-based Prototype (GMDP) for IOD, which explicitly models the distribution relationships of different classes by directly measuring the likelihood of embedding from new and old models into class distribution prototypes. Given a set of input features $X = x_1, x_2, ..., x_D$ from the linear layer of ROI, where each feature $x_i \in \mathbb{R}^D$

represents an object proposal, we aim to learn a set of prototypes $\mathbf{P} = \{\mathbf{p}_1, \mathbf{p}_2, ..., \mathbf{p}_c\}$, where each prototype $\mathbf{p}_c = (\mu_{c,1}, ..., \mu_{c,k}, \sigma_{c,1}, ..., \sigma_{c,k}, \pi_c)$ represents a Gaussian mixture distribution with $K$ components and a mixing coefficient $\pi_c \in \mathbb{R}^K$. Each component is composed of a Gaussian distribution with a mean $\mu_{c,k} \in \mathbb{R}^N$ and a covariance matrices $\sigma_k \in \mathbb{R}^{N \times N}$. For object detection, we compute the likelihood probability of each input feature $x_i$ under each prototype distribution $\mathbf{p}_c$:

$$\mathcal{P}(x_i|\mathbf{p}_c) = \sum_{k=1}^{K} softmax(\pi_c)_k \frac{1}{\sqrt{(2\pi)^N |\sigma_{c,k}|}} \exp\left(-\frac{1}{2}(x_i - \mu_{c,k})^T \sigma_{c,k}^{-1}(x_i - \mu_{c,k})\right), \quad (2)$$

where $K$ is the component number of the Gaussian mixture distribution-based prototype. The $softmax$ function is adopted to ensure that the mixing coefficients sum to 1. Thus the GMDP loss $\mathcal{L}_{GMDP}$ definition for optimizing each class prototype through $x_i$ is as follows:

$$\mathcal{L}_{GMDP} = -\frac{1}{N} \sum_{c=1}^{C} \sum_{i=1}^{N_c} y_{i,c} \log(\mathcal{P}(x_i|\mathbf{p}_c)) + \lambda \sum_{\theta} \|\theta\|^2, \quad (3)$$

where $N$ and $C$ are the number of all features in the batch and total classes. $y_{i,c}$ is the ground-truth label for the $i$-th sample and the $c$-th class. $N_c$ is the number of $c$-th class feature. And $\lambda$ is the regularization weight. $\sum_{\theta} \|\theta\|^2$ is a weight decay term that tends to decrease the magnitude of the weights ($\theta$) of GMDP and penalizes large parameter values to prevent overfitting. $\mathcal{L}_{GMDP}$ measures the difference between the predicted scores and the ground-truth labels. The prediction score of GMDP is computed by taking the maximum likelihood across all prototypes:

$$S_{GMDP}(x_i) = \max_c \mathcal{P}(x_i|\mathbf{p}_c). \quad (4)$$

Finally, we aggregate weighted between the GMDP prediction $S_{GMDP}$ and the IOD detector classification result $S_D$, final classification scores as follows:

$$S_{total}(y|x) = \alpha \cdot S_{GMDP}(y|x) + (1 - \alpha) \cdot S_D(y|x), \quad (5)$$

where $\alpha$ is set to 0.5 to balance the weight between two confidence. During the incremental stage, we transfer old knowledge from the old model to the new model by measuring the divergence between the likelihood probability distributions generated from our GMDP of the old and new models. We can define transfer prototype knowledge loss by Kullback-Leibler (KL) divergence:

$$\mathcal{L}_{tpk} = D_{KL}(\mathcal{P}_o||\mathcal{P}_n) = \sum c \in \mathcal{C}_o \int \mathcal{P}_o(\mathbf{z}|c) \log \frac{\mathcal{P}_o(\mathbf{z}|c)}{\mathcal{P}_n(\mathbf{z}|c)} d\mathbf{z}, \quad (6)$$

where $\mathcal{P}_o(\mathbf{z}|c)$ and $\mathcal{P}_n(\mathbf{z}|c)$ are the likelihood probabilities of the old and new models, respectively, conditioned on the set of old classes $\mathcal{C}_o$. $\mathbf{z}$ is a high-dimensional feature vector extracted from each proposal in the ROI layer. By employing prototypes to represent the feature distributions, our method can adapt to new data without requiring access to the previous task data, thereby overcoming the knowledge shift problem inherent in existing KD-based IOD methods.

**Length scaling progressive learning method**: Introducing GMDP makes network convergence difficult. To address this, we leverage a Length Scaling Progressive Learning (LSPL) method that simplifies the learning process by scaling the length of the learned features and progressively incorporating the GMDP. Formally, if $x \in \mathbb{R}^D$ is the original feature vector, the LSPL reduces its dimensionality to help the model focus on discriminative information, which is expressed as:

$$\mathbf{z} = \sigma(\mathbf{W}x + \mathbf{b}), \quad (7)$$

where $\mathbf{W}$ and $b$ are the weight matrix and bias, respectively. $\sigma(\cdot)$ denotes the activation function. This transforms $x$ into $\mathbf{z} \in \mathbb{R}^d$, where $d < D$, retaining crucial feature information. Besides, we adopted an initial training phase to relieve cold start, first conducting N training steps and then employing our GMDP to construct high-dimensional prototypes.

### 3.3 DYNAMIC ADAPTIVE PROTOTYPE OPTIMIZATION STRATEGY

To enhance the plasticity of class prototypes, we present a Dynamic Adaptive Prototype Optimization (DAPO) strategy for IOD during the incremental process, which comprises three core principles aimed at enhancing the adaptability and effectiveness of GMDP modeling.

**Inter-Class Mean Dispersal**: The first principle emphasizes maximizing the separation between the means of different classes, thereby promoting greater dissimilarity among prototypes. By increasing inter-class mean dispersion of GMDP, the distance between distinct prototypes is maximized, facilitating better discrimination between classes. The Inter-Class Mean Dispersal (IMD) loss measures the distance between means of different classes of GMDP:

$$\mathcal{L}_{IMD} = \sum_{i=1}^{C} \sum_{j=i+1}^{C} \frac{1}{\|\mu_i - \mu_j\|_2 + \epsilon},$$

(8)

where $C$ is the number of classes, $\mu_i$ is the mean of class $i$, and $\epsilon$ is a small constant.

**Intra-Class Component Cohesion**: The second principle focuses on minimizing the distances between components of GMDP within the same class. This ensures that the components of a class's GMDP are closely clustered, enhancing the coherence and representativeness of the class distribution prototype. The Intra-Class Component Cohesion (ICC) loss measures the distance between means of different components within the same class:

$$\mathcal{L}_{ICC} = \sum_{i=1}^{C} \sum_{j=1}^{K} \sum_{k=j+1}^{K} \|\mu_{i,j} - \mu_{i,k}\|_2,$$

(9)

where $M$ is the number of components, and $\mu_{i,j}$ is the mean of $j$-th component of $i$-th class.

**Minimization of Prototype Variance**: The third principle aims to mitigate the variance of GMDP across disparate classes. By minimizing intra-class prototype variance, the distributions become more compact, providing a more structured and discriminative feature space that facilitates superior performance in subsequent tasks. The Minimization of Prototype Variance (MPV) loss aggregates the variances of all components as follows:

$$\mathcal{L}_{MPV} = \sum_{i=1}^{C} \sum_{j=1}^{K} \sigma_{i,j},$$

(10)

where $\sigma_{i,j}$ is the variance of the $j$-th component of class $i$. By incorporating these principles into the optimization process, our approach enables prototypes to dynamically adapt to task requirements.

Table 1: mAP@0.5% results on the two-stage incremental setting on Pascal VOC. The best and the second best results are highlighted in **bold** and underlined. Two methods equipped with our idea consistently outperform baselines by a large margin and achieve state-of-the-art results compared with existing methods in different incremental settings.

| Method | | 19-1 | | | 15-5 | | | 10-10 | | | 5-15 | | |
|---|---|---|---|---|---|---|---|---|---|---|---|---|---|
| | | 1-19 | 20 | 1-20 | 1-15 | 16-20 | 1-20 | 1-10 | 11-20 | 1-20 | 1-5 | 6-20 | 1-20 |
| Joint Training | | 70.1 | 75.7 | 74.3 | 76.4 | 67.8 | 74.3 | 75.5 | 73.0 | 74.3 | 70.1 | 75.7 | 74.3 |
| Fine-tuning | | 11.8 | 64.7 | 14.4 | 15.9 | 54.2 | 25.5 | 2.6 | 63.4 | 32.9 | 6.9 | 63.1 | 49.1 |
| Faster ILOD [47] | *PRL 20* | 70.9 | 63.2 | 70.6 | 73.1 | 57.3 | 69.2 | 70.3 | 53.0 | 61.7 | 62.0 | 37.1 | 43.3 |
| MVC [66] | *PR 22* | 70.2 | 60.6 | 69.7 | 69.4 | 57.9 | 66.5 | 66.2 | 66.0 | 66.1 | - | - | - |
| ORE [24] | *CVPR 21* | 69.4 | 60.1 | 68.9 | 71.8 | 58.7 | 68.5 | 60.4 | 68.8 | 64.6 | - | - | - |
| Meta-ILOD [25] | *TPAMI 21* | 70.9 | 57.6 | 70.2 | 71.7 | 55.9 | 67.8 | 68.4 | 64.3 | 66.3 | - | - | - |
| OW-DETR [17] | *CVPR 22* | 70.2 | 62.0 | 69.8 | 72.2 | 59.8 | 69.1 | 63.5 | 67.9 | 65.7 | - | - | - |
| MMA [5] | *CVPR 22* | 70.9 | 62.9 | 70.5 | 72.7 | 60.6 | 69.7 | 69.8 | 63.9 | 66.8 | 66.8 | 57.2 | 59.6 |
| CAT [40] | *CVPR 23* | 74.5 | 61.1 | 73.8 | **76.5** | 59.3 | 72.2 | 70.0 | 67.4 | 67.7 | - | - | - |
| Wu*et al.* [63] | *TIP 24* | 72.5 | 61.4 | 71.9 | 73.6 | 62.0 | 70.7 | 70.3 | 68.7 | 69.5 | - | - | - |
| ILOD [57] | *CVPR 17* | 69.8 | 64.5 | 69.6 | 72.5 | 58.5 | 68.9 | 69.8 | 53.7 | 61.7 | 61.0 | 37.3 | 43.2 |
| **GMDP-ILOD** | *Ours* | 74.2 | 67.9 | 73.9 | 74.6 | 63.5 | 71.8 | 71.9 | 69.7 | 70.8 | 65.2 | 60.5 | 61.7 |
| ABR [38] | *ICCV 23* | 71.0 | 69.7 | 70.9 | 73.0 | 65.1 | 71.0 | 71.2 | 72.8 | 72.0 | 64.7 | 71.0 | 69.4 |
| **GMDP-ABR** | *Ours* | **74.8** | **70.1** | **74.6** | 75.8 | **65.5** | **73.2** | **72.1** | **73.2** | **72.7** | **67.1** | **71.9** | **70.7** |

# 4 EXPERIMENTS

## 4.1 DATASETS AND EVALUATION

For a fair comparison, we followed the previous work [57; 38] to adopt two widely used incremental object detection (IOD) datasets to evaluate the effectiveness of our method: PASCAL VOC

Table 2: mAP@0.5% results on the multiple-stage incremental setting on Pascal VOC. The best and the second best results are highlighted in **bold** and underlined. Two baselines with our idea are able to forget less about the knowledge of previous tasks and learn better new information.

| Method | | 10-5 (3 tasks) | | | 5-5 (4 tasks) | | | 10-2 (6 tasks) | | | 15-1 (6 tasks) | | | 10-1 (11 tasks) | | |
|---|---|---|---|---|---|---|---|---|---|---|---|---|---|---|---|---|
| | | 1-10 | 11-20 | 1-20 | 1-5 | 6-20 | 1-20 | 1-10 | 11-20 | 1-20 | 1-15 | 16-20 | 1-20 | 1-10 | 11-20 | 1-20 |
| Joint Training | | 75.5 | 73.0 | 74.3 | 70.1 | 75.7 | 74.3 | 75.5 | 73.0 | 74.3 | 76.4 | 67.8 | 74.3 | 75.5 | 73.0 | 74.3 |
| Fine-tuning | | 5.3 | 30.6 | 18.0 | 0.5 | 18.3 | 13.8 | 3.79 | 13.6 | 8.7 | 0.0 | 10.47 | 5.3 | 0.0 | 5.1 | 2.55 |
| Faster ILOD [47] | *PRL 20* | 68.3 | 57.9 | 63.1 | 55.7 | 16.0 | 25.9 | 64.2 | 48.6 | 56.4 | 66.9 | 44.5 | 61.3 | 53.5 | 41.0 | 47.3 |
| MMA [5] | *CVPR 22* | 67.4 | 60.5 | 64.0 | 62.3 | 31.2 | 38.9 | 65.7 | 52.5 | 59.1 | 67.2 | 47.8 | 62.3 | 57.9 | 44.6 | 51.2 |
| ILOD [57] | *CVPR 17* | 67.2 | 59.4 | 63.3 | 58.5 | 15.6 | 26.3 | 62.1 | 49.8 | 55.9 | 65.6 | 47.6 | 60.2 | 52.9 | 41.5 | 47.2 |
| **GMDP-ILOD** | *Ours* | 68.1 | 62.3 | 65.2 | 61.1 | 35.8 | 42.1 | 64.2 | 53.3 | 58.8 | 66.8 | 50.4 | 62.7 | 56.2 | 50.7 | 53.5 |
| ABR [38] | *ICCV 23* | 68.7 | 67.1 | 67.9 | 64.7 | 56.4 | 58.4 | 67.0 | 58.1 | 62.6 | 68.7 | 56.7 | 65.7 | 62.0 | 55.7 | 58.9 |
| **GMDP-ABR** | *Ours* | 69.9 | 67.8 | 68.9 | 66.3 | 59.3 | 61.1 | 67.6 | 58.9 | 63.3 | 69.5 | 58.9 | 66.9 | 63.3 | 57.5 | 60.4 |

2007 [12] and MS-COCO [32]. PASCAL VOC 2007 comprises 9,963 images across 20 categories, with 50% of them as training and validation sets. The remaining half is reserved for testing. The evaluation metric employs mean Average Precision (mAP) using a 0.5 IoU threshold. MS-COCO is a challenging dataset containing 80 object categories. We report the COCO-style Average Precision (AP) at different IoU ranging from 0.5 to 0.95 (mAP@[50:95]), 0.50 IoU (mAP@50), and 0.75 IoU (mAP@75) as the evaluation.

## 4.2 IMPLEMENTATION DETAILS

In this paper, our experiments are conducted upon four robust baselines: ILOD [57], ERD [13], CL-DETR [36] and ABR [38]. Comprehensive results of Transformer baselines ERD and CL-DETR are provided in the Appendix. Following prior work [57; 13; 38], we employ the ResNet-50 [20] as the backbone, initializing ILOD and ABR baselines with pre-trained models from ImageNet [53]. Fitting validation based on each feature length in Section 3.1 confirms that the maximum number of components for the Gaussian Mixture Distribution is 2. Adjusting the weights of the components can also be applied to a single Gaussian Mixture Distribution. All experiments are carried out on 4 Nvidia GeForce RTX 3090 GPUs with a batch size of 16 implemented by PyTorch.

## 4.3 QUANTITATIVE EVALUATION

### 4.3.1 RESULTS ON PASCAL VOC.

We partitioned the datasets into distinct incremental task sequences in class-incremental object detection. PASCAL VOC 2007 was split into two-stage and multi-stage incremental task settings. For the two-stage incremental setting, tasks are divided into 19-1, 15-5, 10-10, and 5-15, incrementing by 1, 5, 10, and 15 classes, respectively. The multi-stage incremental setting divides tasks into 10-5, 5-5, 10-2, 15-1, and 10-1, comprising 3, 4, 6, 6, and 11 tasks, respectively.

**Two-stage increments.** In Table 1, we compare our method to state-of-the-art approaches. We observed that employing the fine-tuning method across all settings led to catastrophic forgetting [45]. This resulted in significant hindrances in learning new tasks due to forgetting prior knowledge, leading to knowledge confusion. Compared to ILOD [57], our approach demonstrates stable improvements across both old and new tasks, with a notable 15% improvement in the 10-10 setting and a 43% improvement in the 15-5 setting. GMDP-ILOD achieves competitive performance compared with recent state-of-the-art methods under the classic IOD method from CVPR in 2017. Based on advanced ABR [38], our method outperforms the baseline by a large margin and achieves state-of-the-art performance, further validating the effectiveness of our method.

**Multi-stage increments.** To demonstrate the performance of our IOD method, we also show the performance of multi-step incremental learning on Pascal VOC as shown in Table 2. It can be observed that fine-tuning exhibits a more severe catastrophic forgetting over multiple steps, where the forgetting of old knowledge also leads to the limited performance of new classes. Our IOD method surpasses the two baselines by a large margin and performs well on each incremental step. Moreover, our method improves both old and new class performance against the baseline model. This is attributed to the feature explicit modeling for more accurate GMDP, capturing the essential features according to their distribution characteristics.

Table 3: mAP results on MS COCO 2017 at different IoU, where the best among columns in **bold**.

| Method | 40-40 mAP@ | | | 70-10 mAP@ | | |
|---|---|---|---|---|---|---|
| | $[50:95]$ | 50 | 75 | $[50:95]$ | 50 | 75 |
| Joint Training | 35.9 | 60.5 | 38.0 | 35.9 | 60.5 | 38.0 |
| Fine-tuning | 19.0 | 31.2 | 20.4 | 5.6 | 8.6 | 6.2 |
| Faster ILOD [47] | 20.6 | 40.1 | - | 21.3 | 39.9 | - |
| MMA [5] | 33.0 | 56.6 | 34.6 | 30.2 | 52.1 | 31.5 |
| ABR [38] | 34.5 | 57.8 | 35.2 | 31.1 | 52.9 | 32.7 |
| **GMDP-ABR** | **36.8** | **59.6** | **36.7** | **32.5** | **53.8** | **33.9** |

Table 4: Ablation study of different components of our method for IOD on Pascal-VOC 2007. After training with the GMDP, our model achieves a significant improvement over the baseline, especially in previous tasks. The performance of new tasks could be further enhanced by DAPO strategy.

| Baseline | GMDP | DAPO | | | 19-1 | | | 15-5 | | | 10-10 | | | Speed |
|---|---|---|---|---|---|---|---|---|---|---|---|---|---|---|
| | | $\mathcal{L}_{IMD}$ | $\mathcal{L}_{ICC}$ | $\mathcal{L}_{MPV}$ | 1-19 | 20 | 1-20 | 1-15 | 16-20 | 1-20 | 1-10 | 11-20 | 1-20 | |
| | | Joint Training | | | 70.1 | 75.7 | 74.3 | 76.4 | 67.8 | 74.3 | 75.5 | 73.0 | 74.3 | - |
| | | Fine-tuning | | | 11.8 | 64.7 | 14.4 | 15.9 | 54.2 | 25.5 | 2.6 | 63.4 | 32.9 | - |
| ✓ | | | | | 69.8 | 64.5 | 69.6 | 72.5 | 58.5 | 68.9 | 69.8 | 53.7 | 61.7 | 27.2fps |
| ✓ | ✓ | | | | 73.8 | 65.2 | 73.4 | 73.3 | 60.2 | 70.0 | **72.0** | 66.3 | 69.2 | 25.3fps |
| ✓ | ✓ | ✓ | | | 74.0 | 66.9 | 73.6 | 73.9 | 62.1 | 71.0 | 71.9 | 69.1 | 70.5 | 25.3fps |
| ✓ | ✓ | | ✓ | | 73.7 | 65.7 | 73.3 | 73.7 | 61.8 | 70.7 | 72.1 | 67.2 | 69.5 | 25.3fps |
| ✓ | ✓ | | | ✓ | 73.9 | 67.1 | 73.6 | 74.0 | 62.3 | 71.1 | 71.8 | 68.9 | 70.4 | 25.3fps |
| ✓ | ✓ | ✓ | ✓ | ✓ | **74.2** | **67.9** | **73.9** | **74.6** | **63.5** | **71.8** | 71.9 | **69.7** | **70.8** | 25.3fps |

### 4.3.2 RESULTS ON MS-COCO.

As indicated in Table 3, we conduct the experiments on MS-COCO to compare the proposed GMDP with existing methods under 40-40 and 70-10 incremental settings. Fine-tuning also suffers from catastrophic forgetting across all settings. The performance of our method exceeds its baseline. Under 40-40 and 70-10 settings, our method achieves 7%, 5% performance gain compared with ABR [38] on mAP@[50:95] respectively. After applying the GMDP and DAPO strategy, our method outperforms all other methods significantly. The results show that our prototype modeling method for IOD performs better on the more difficult MS-COCO dataset.

### 4.4 ABLATION STUDIES AND DISCUSSION

To prove the effectiveness of each component, we perform ablation studies as shown in Table 4. All experiments are conducted on PASCAL VOC 2007 with ILOD [57] as the base detector. We conducted experiments with GMDP based on ILOD, observing significant performance improvements (Table 4 row 3 vs row 4) attributed to GMDP's ability to explicitly model distributional characteristics, resulting in improved IOD performance and generalization capabilities. Importantly, the model retained valuable information according to their distribution characteristics from previous tasks, thereby mitigating catastrophic forgetting. We also evaluated the impact of the three loss functions, $\mathcal{L}_{IMD}$, $\mathcal{L}_{ICC}$, and $\mathcal{L}_{MPV}$ of DAPO on model performance. Adopting the complete DAPO strategy notably enhanced adaptability and performance on new tasks, indicating its efficacy in facilitating knowledge transfer and adaptation by dynamically optimizing GMDP.

In Figure 6, we compare the effectiveness of various training methods for IOD through visualization of feature distributions. Joint Training illustrates the feature distributions for different classes are well-separated, maximizing the model's performance. The first 5 classes of Fine-tuning are randomly scattered and mixed with other classes, resulting in catastrophic forgetting. The Statistical Gaussian Distribution-based Prototype (SGDP) mitigates some aspects of catastrophic forgetting, but a notable overlap exists between old and new tasks with many outliers. This indicates that SGDP is ineffective in preserving prior knowledge under complex high-dimensional feature spaces. Our GMDP effectively maintains the separation of different classes, with minimal overlap. This demonstrates its capability to preserve prior knowledge and integrate new tasks seamlessly, achieving superior IOD performance.

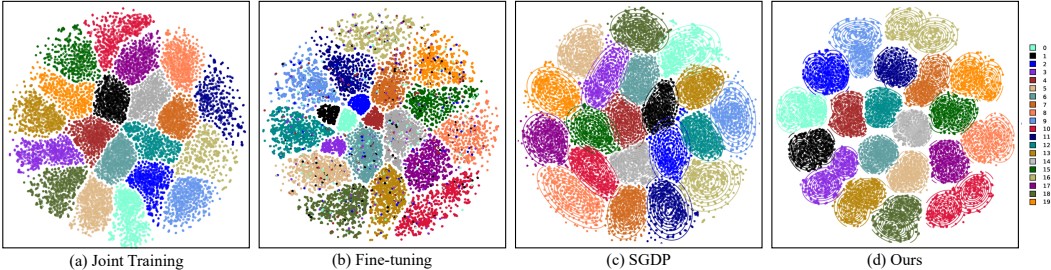

|              |              |           |            |           |
| (a) Joint Training | (b) Fine-tuning | (c) SGDP | (d) Ours |

Figure 6: The t-SNE visualization results with different learning methods under a 5-15 task setting. (a) Joint Training is trained on all tasks simultaneously, clearly separating classes. (b) Fine-tuning method is sequentially trained on new tasks, leading to severe catastrophic forgetting. (c) SGDP mitigates some forgetting but still shows significant overlap between old and new task features, along with many outliers, indicating inadequate retention of previous knowledge. (d) Clusters generated by our method are more compact and can be well separated from each other.

Table 5: (Left) Analysis of different numbers of Gaussian components in GMDP. (Right) Visualization of decision boundary and the t-SNE for different classes of GMDP with 4 components.

| Num of       |      | 19-1 |      |      | 15-5  |      |
| Component    | 1-19 | 20   | 1-20 | 1-15 | 16-20 | 1-20 |
|--------------|------|------|------|------|-------|------|
| 0            | 69.8 | 64.5 | 69.6 | 72.5 | 58.5  | 68.9 |
| 1            | 73.8 | 67.2 | 73.5 | 73.5 | 63.1  | 70.9 |
| 2            | **74.2** | **67.9** | **73.9** | **74.6** | **63.5** | **71.8** |
| 3            | 73.5 | 66.9 | 73.2 | 73.1 | 62.7  | 70.5 |
| 4            | 71.3 | 65.9 | 71.0 | 71.7 | 61.1  | 69.0 |

(a) The decision boundary diagram with 4 components.     (b) The t-SNE visualization with 4 components.

In addition, we have carefully explored the impact of increasing the number of components in GMDP under the PASCAL VOC dataset in Table 5 (Left). We can see that the GMDP with 2 components consistently improves performance compared to 1 component, allowing for a more accurate representation of higher-dimensional prototypes. Although theoretically, more components can better fit complex feature distributions (The decision boundary of the 4 components is more accurate in Table 5 (Right (a)), in practice, this poses significant challenges to the network's fitting capability. 4 components result in a decrease in the detection performance of the proposed method, as evidenced by the t-SNE visualization in Table 5 (Right (b)), which displays many outliers. This difficulty in the convergence of more components is a critical issue. Our Length Scaling Progressive Learning method addresses these challenges by simplifying the learning process, enabling optimal performance under 2 components setting. Exploring the potential of adopting more components is an avenue for our future work.

### 4.5 LIMITATION

The Introduced GMDP increases the model's complexity, making end-to-end training more challenging and requiring an initial training phase before integrating our method. This step makes it easier for the model to converge. Meanwhile, as shown in the last column of Table 4, we also evaluate testing speed, where our method is nearly 7% slower than the baseline (row 3 vs row 8). Despite this, our method demonstrates significant performance improvements under 4 baselines for all tasks.

## 5 CONCLUSION

In this paper, we propose a novel prototype learning approach for KD-based IOD to alleviate the knowledge shift, which enables a more flexible, accurate, and fine-grained representation of feature distributions without keeping any previous task data. A Gaussian Mixture Distribution-based Prototype (GMDP) is introduced to explicitly model the distribution relationships between different classes suitable for more complex feature spaces. To enhance the plasticity of class prototypes, we present a Dynamic Adaptive Prototype Optimization Strategy (DAPO) strategy for IOD during the incremental process, which could enhance the adaptability and effectiveness of GMDP modeling. The proposed method can be readily integrated into existing IOD methods to enhance performance further. Methods with our idea on the PASCAL VOC and MS COCO consistently outperform four baselines significantly and achieve state-of-the-art results in different incremental settings.

## ACKNOWLEDGEMENT

This work is supported by the National Natural Science Foundation of China (NSFC) grants 62176100, U24B20139, and 62376011. The computation is completed in the HPC Platform of Huazhong University of Science and Technology.

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

# A    APPENDIX

## A.1    IMPLEMENTATION DETAILS

Our method is based on ILOD [57] with the SGD optimizer, setting the initial learning rate to 0.001, decaying to 0.0001 after 30,000 iterations, and incorporating a momentum of 0.9. During the incremental phase, the learning rate is adjusted to 0.0001. Following the original paper, other settings are the same as the original paper. Our GMDP-ERD based on ERD [13] employs the GFLV1 [29] detector with ResNet-50 as the backbone. Following the original paper, we adopt the same settings to train the detector. 12 epochs train the method for each incremental step. Our method based on CL-DETR [36] adopts a pre-trained ResNet-50 as the backbone. We train the model with 50 epochs under different incremental stages. Our GMDP-ABR follows the baseline ABR [38] train model with the SGD optimizer, weight decay of 0.0001, and momentum of 0.9. The first stage has a learning rate of 0.005, and the subsequent stages have a learning rate of 0.002. For tasks involving incremental addition of 5 or 10 classes, we employed 15,000 iterations, while for the inclusion of 1 or 2 new classes, a regimen of 5,000 iterations was applied.

In summary, the total loss ($\mathcal{L}_{total}$) of the training model under the incremental stage is the weighted sum of the original baseline loss $\mathcal{L}_{Det}$, GMDP loss $\mathcal{L}_{GMDP}$, transferring prototype knowledge loss $\mathcal{L}_{tpk}$, and DAPO strategy loss. The $\mathcal{L}_{logit}$ and $\mathcal{L}_{reg}$ in Figure 5 are part of $\mathcal{L}_{Det}$, represent the loss functions in the baseline model, corresponding to the distillation loss for class classification and bounding box regression, respectively. The DAPO strategy loss consists of three parts: Inter-Class Mean Dispersal loss $\mathcal{L}_{IMD}$, Intra-Class Component Cohesion loss $\mathcal{L}_{ICC}$, and Minimization of Prototype Variance loss $\mathcal{L}_{MPV}$. The total loss of the incremental object detection network is as follows:

$$\mathcal{L}_{total} = \mathcal{L}_{Det} + \lambda_1 \mathcal{L}_{GMDP} + \lambda_2 \mathcal{L}_{tpk} + \lambda_3 (\mathcal{L}_{IMD} + \mathcal{L}_{ICC} + \mathcal{L}_{MPV}), \tag{11}$$

where the parameters $\lambda_1 = 0.05$, $\lambda_2 = 0.1$, and $\lambda_3 = 0.05$ represent the weight of different losses, thereby regulating the tradeoff between them. These parameters primarily aim to standardize the range of loss values within the same range rather than implementing intricate adjustments or adding bells and whistles. Adopting GMDP introduced additional parameters, leading to slower network convergence. To address this issue, we leverage a Length Scaling Progressive Learning (LSPL) method that enhances the network convergence speed by reducing the dimensionality of the detection features to 256. Sufficient experimental results indicate that despite reducing feature dimensionality, our detection performance remained unaffected, and we maintained the same network iteration settings as the original baseline model. Besides, during the initial stage of learning new classes, the feature distribution is inherently unstable. Thus, significant challenges have been posed for the learning and convergence of GMDP, due to the lack of stable feature dependency relationships. To address this issue, we adopted an initial training phase, first conducting 1000 training steps and then employing our GMDP to construct the Gaussian distribution prototype.

## A.2    ADDITIONAL EXPREIMENTS

In this section, we conducted IOD experiments on the Ms-COCO 2017 dataset based on two different baselines, ERD [13] and CL-DETR [36] to evaluate our method. As shown in Table 6, our experiments are conducted on two-stage task settings, adding 40 and 10 categories in 40-40 and 70-10 setups, respectively. Each experiment was conducted three times, with the classes and data order randomized in each phase. Our approach consistently outperforms both two baselines across various IOD settings, demonstrating significant improvements. Furthermore, our results are competitive with the current state-of-the-art (SOTA) methods, indicating the effectiveness and robustness of our proposed method in diverse IOD settings. This improvement can be attributed to GMDP's ability to explicitly model the distributional characteristics of detection features, allowing for more accurate and discriminative representations. By dynamically optimizing prototype representations, the DAPO strategy enables the model to quickly adapt to new task requirements, improving IOD performance and generalization capabilities.

We also validated the effectiveness of our method on the Ms-COCO dataset under the multi-stage incremental setting. Table 7 displays our results with additional multi-stage experiments (e.g., 40+10×4, 40+20×2) to provide a comprehensive demonstration of our method's performance under various incremental settings. , our method achieves consistent performance improvement compared

Table 6: mAP results on MS-COCO 2017 at different IoU. We run experiments for three different classes and data orders.

| Setting | Method | | Baseline | AP | AP50 | AP75 | APS | APM | APL |
|---|---|---|---|---|---|---|---|---|---|
| 70+10 | LwF [31] | *TPAMI 18* | GFLV1 | 7.1 | 12.4 | 7.0 | 4.8 | 9.5 | 10.0 |
| | ERD [13] | *CVPR 22* | GFLV1 | 34.9 | 51.9 | 37.4 | 18.7 | 38.8 | 45.5 |
| | GMDP-ERD | *Ours* | GFLV1 | $36.4_{\pm 0.3}$ | $53.2_{\pm 0.5}$ | $39.1_{\pm 0.2}$ | $19.4_{\pm 0.3}$ | $39.7_{\pm 0.1}$ | $46.9_{\pm 0.5}$ |
| | CL-DETR [36] | *CVPR 23* | UP-DETR | 37.6 | 56.5 | 39.4 | 20.5 | 39.1 | 49.9 |
| | LwF [31] | *TPAMI 18* | Deform-DETR | 24.5 | 36.6 | 26.7 | 12.4 | 28.2 | 35.2 |
| | iCaRL [49] | *CVPR 17* | Deform-DETR | 35.9 | 52.5 | 39.2 | 19.1 | 39.4 | 48.6 |
| | CL-DETR [36] | *CVPR 23* | Deform-DETR | 40.1 | 57.8 | 43.7 | 23.2 | 43.2 | 52.1 |
| | GMDP-CL-DETR | *Ours* | Deform-DETR | $\mathbf{42.6_{\pm 0.5}}$ | $\mathbf{60.4_{\pm 0.4}}$ | $\mathbf{45.2_{\pm 0.3}}$ | $\mathbf{24.1_{\pm 0.2}}$ | $\mathbf{45.0_{\pm 0.2}}$ | $\mathbf{55.6_{\pm 0.4}}$ |
| 40+40 | LwF [31] | *TPAMI 18* | GFLV1 | 17.2 | 25.4 | 18.6 | 7.9 | 18.4 | 24.3 |
| | ERD [13] | *CVPR 22* | GFLV1 | 36.9 | 54.5 | 39.6 | 21.3 | 40.4 | 47.5 |
| | GMDP-ERD | *Ours* | GFLV1 | $38.4_{\pm 0.1}$ | $55.1_{\pm 0.3}$ | $40.3_{\pm 0.1}$ | $22.9_{\pm 0.3}$ | $41.7_{\pm 0.2}$ | $49.8_{\pm 0.4}$ |
| | CL-DETR [36] | *CVPR 23* | UP-DETR | 37.0 | 56.2 | 39.1 | 20.9 | 38.9 | 49.2 |
| | LwF [31] | *TPAMI 18* | Deform-DETR | 23.9 | 41.5 | 25.0 | 12.0 | 26.4 | 33.0 |
| | iCaRL [49] | *CVPR 17* | Deform-DETR | 33.4 | 52.0 | 36.0 | 18.0 | 36.4 | 45.5 |
| | CL-DETR [36] | *CVPR 23* | Deform-DETR | 37.5 | 55.1 | 40.3 | 20.9 | 40.8 | 50.7 |
| | GMDP-CL-DETR | *Ours* | Deform-DETR | $\mathbf{40.8_{\pm 0.4}}$ | $\mathbf{58.5_{\pm 0.2}}$ | $\mathbf{43.3_{\pm 0.2}}$ | $\mathbf{23.6_{\pm 0.5}}$ | $\mathbf{43.5_{\pm 0.3}}$ | $\mathbf{53.5_{\pm 0.4}}$ |

to the baseline and performs well on each incremental step. This is attributed to the feature explicit modeling for more accurate GMDP, capturing the essential features according to their distribution characteristics. It is noteworthy that due to the time limitation, we can not implement our idea of transformer-based methods. only Faster-RCNN-based methods are used for fair comparisons.

Table 7: GMDP result (AP/AP50) on the multiple-stage incremental setting on MS COCO. We reproduced the results of ABR* with the same setting.

| Method | T1 (1-40) | 40+10×4 | | | | 40+20×2 | |
|---|---|---|---|---|---|---|---|
| | | T2 (40-50) | T3 (50-60) | T4 (60-70) | T5 (70-80) | T2 (40-60) | T3 (60-80) |
| ERD [13] | 45.7/66.3 | 36.4/53.9 | 30.8/46.7 | 26.2/39.9 | 20.7/31.8 | 36.7/54.6 | 32.4/48.6 |
| RILOD [27] | 45.7/66.3 | 25.4/38.9 | 11.2/17.3 | 10.5/15.6 | 8.4/12.5 | 27.8/42.8 | 15.8/4.0 |
| SID [48] | 45.7/66.3 | 34.6/52.1 | 24.1/38.0 | 14.6/23.0 | 12.6/23.3 | 34.0/51.8 | 23.8/36.5 |
| ABR* [38] | 45.7/66.3 | 37.9/56.3 | 35.5/52.4 | 31.5/47.6 | 28.5/43.5 | 38.0/57.5 | 34.1/50.3 |
| GMDP-ABR | **46.3/67.5** | **39.2/59.1** | **36.8/55.4** | **33.2/49.3** | **29.7/45.1** | **39.6/60.2** | **35.7/52.9** |

Table 8: Analysis of different number of Gaussian component in GMDP under multiple-stage incremental settings.

| Num of Component | 10-5 (3 tasks) | | | 5-5(3 tasks) | | | 15-1(6 tasks) | | |
|---|---|---|---|---|---|---|---|---|---|
| | 1-10 | 11-20 | 1-20 | 1-5 | 6-20 | 1-20 | 1-15 | 16-20 | 1-20 |
| 0 | 68.7 | 67.1 | 67.9 | 64.7 | 56.4 | 58.4 | 68.7 | 56.7 | 65.7 |
| 1 | 68.9 | 67.5 | 68.2 | 66.0 | 58.7 | 60.5 | 69.1 | 58.3 | 66.4 |
| 2 | **69.9** | **67.8** | **68.9** | **66.3** | **59.3** | **61.1** | **69.5** | **58.9** | **66.9** |
| 3 | 69.4 | 67.2 | 68.3 | 65.7 | 58.5 | 60.3 | 68.9 | 58.1 | 66.2 |
| 4 | 68.6 | 67.0 | 67.8 | 64.9 | 57.2 | 59.1 | 68.3 | 57.0 | 65.5 |

We have carefully analyzed the influence of setting different numbers of Gaussian components in GMDP under multiple-stage incremental settings. Illustrated in 8 different numbers of Gaussian components are used in the experiment on the PASCAL VOC dataset. We can see that the GMDP with 2 components under multiple-stage incremental settings consistently improves performance compared to 1 component, with results similar to 2-stage incremental settings. The proposed method is insensitive and often achieves performance improvement when the threshold changes from 1 to 3.

Although theoretically, more components can better fit complex feature distributions, in practice, this poses significant challenges to the network's fitting capability. 4 components result in a decrease in the detection performance of the proposed method. This difficulty in the convergence of more components is a critical issue. Our Dimension Scaling Progressive Learning method addresses these challenges by simplifying the learning process, enabling optimal performance under 2 components

setting. The above results also demonstrate the effectiveness and consistency of our method in two-stage and multi-stage incremental settings.

