# OpenReview forum: "High-dimension Prototype is a Better Incremental Object Detection Learner"
_ICLR.cc/2025/Conference — ICLR 2025 Poster_

### Official Review · Reviewer_vCwD · 2024-10-30

**Soundness:** 3
**Presentation:** 1
**Contribution:** 3
**Rating:** 8
**Confidence:** 4

**Summary:**

This paper introduces a higher-dimension prototype learning method that incorporates GMDP to address issues such as knowledge shift present in KD-based IOD methods. Additionally, a DAPO strategy is designed to enhance the adaptability and effectiveness of GMDP modeling. The authors conducted experiments on PASCAL VOC and MS-COCO, demonstrating the validity of the proposed method.

**Strengths:**

1. This paper proposes a novel approach by introducing a Gaussian Mixture Distribution-based Prototype(GMDP) to alleviate the knowledge shift.
2. GMDP can be readily integrated into existing IOD methods for better performance.
3. Experiments show that this method achieves SOTA results on PASCAL VOC and MS COCO.
4. The motivation is clearly articulated, and it includes figures that illustrate the issues with current IOD methods as well as the differences in feature spaces for classification and detection tasks.

**Weaknesses:**

1. The quality and clarity of the figures need further improvement. The legend for Figure 1 is inconsistent, and Figure 3 is blurry and has a low resolution.
2. Some descriptions need to be more precise and clear. For example, in line 303, "the old class C_o" should refer to a "class set". The variable z in Equation (6) has not been introduced; it should refer to proposals generated from the ROI layer. Additionally, how are the means and variances of the K components of GMD calculated? Are they based on the features x? For incremental learning tasks, is the computation of these means and variances confirmed before or after training?
3. As mentioned in line 515, "this poses significant challenges to the network’s fitting capability.", it lacks an experiment on the selection of the number of Gaussian componenets for larger models.

**Questions:**

Some descriptions need to be more precise and clear. For example, in line 303, "the old class C_o" should refer to a "class set". The variable z in Equation (6) has not been introduced; it should refer to proposals generated from the ROI layer. Additionally, how are the means and variances of the K components of GMD calculated? Are they based on the features x? Is the computation of these means and variances confirmed before or after training for incremental learning tasks?

---

### Official Review · Reviewer_sEU9 · 2024-11-02

**Soundness:** 3
**Presentation:** 4
**Contribution:** 3
**Rating:** 8
**Confidence:** 3

**Summary:**

The paper focuses on incremental object detection (IOD) and aims to address knowledge shift using enhanced high-dimensional prototype learning. Specifically, the authors propose a Gaussian Mixture Distribution-based Prototype (GMDP) to model the complex feature distribution based on both new and old models. To facilitate GMDP modeling, they further introduce a Dynamic Adaptive Prototype Optimization Strategy (DAPO). The proposed method can be readily integrated into existing IOD approaches, consistently improving their results.

**Strengths:**

1. The investigation into why accurate modeling of the complex distribution for IOD is necessary and compelling.
2. The experiments are convincing and comprehensive.
3. The paper is easy to follow, and the visualizations are helpful, making it easy for readers to grasp the main idea.

**Weaknesses:**

1. Lack of discussion regarding method complexity and training time.
2. As the core contribution involves adopting Gaussian Mixture Distribution, it would be beneficial to conduct ablation studies on the number of components in both two-stage and multiple-stage incremental settings.
3. Minor: some of the best results are not boldfaced in Table 4.

**Questions:**

1. Do all classes share the same number of components in GMDP? Are there insights into how to determine the number of components for different classes? Intuitively, a unimodal Gaussian distribution may be sufficient for easily distinguished classes, while a multiple-component Gaussian distribution is more suitable for harder classes.
2. What are the weights of the losses in the DAPO? How are these values determined across different settings?

---

### Official Review · Reviewer_A3Cy · 2024-11-06

**Soundness:** 2
**Presentation:** 3
**Contribution:** 2
**Rating:** 5
**Confidence:** 5

**Summary:**

This paper introduces a new higher-dimension-prototype learning approach for knowledge distillation-based incremental object detection, which uses Gaussian mixture distributions to model the feature distributions of classes more effectively. This approach helps address the issue of knowledge shift caused by the introduction of new classes without access to previous task data. Dynamic Adaptive Prototype Optimization strategy improves the separation of class features and enhances the plasticity of class prototypes. Comprehensive experimental results support the efficacy of the proposed method.

**Strengths:**

1. This paper leverages the flexibility and expressiveness of Gaussian mixtures to capture the complex distributions of object features in high-dimensional spaces, alleviating the knowledge shift caused by the lack of previous data.

2. The proposed Dynamic Adaptive Prototype Optimization (DAPO) strategy is a well-reasoned enhancement that improves the plasticity and stability of class prototypes. By optimizing the separation and cohesion of class features, the strategy provides a robust mechanism for managing the intricacies of incremental learning.

3. The paper is well-organized and clearly written, which facilitates readers' understanding of the proposed approach.

4. The method is validated through extensive experiments, demonstrating consistent superiority over existing state-of-the-art methods.

**Weaknesses:**

1. Gaussian Mixture Distribution-based prototypes have been previously explored in class incremental learning[1-4]. Specifically, [1] highlights that embedding distributions in feature space are not convex or isotropic in practice. Thus, the author’s emphasis on the complexity of detection features compared to classification features may not provide substantial new insights. The innovation level is thus moderate.

2. This method is not specifically tailored for incremental object detection (IOD) tasks, which appears more suited for general class-incremental learning for modeling class distributions.

3. The integration of GMDP and DAPO at the base stage of the model may inherently enhance the detector’s feature representation capacity. Given that the upper bound of Faster R-CNN has been exceeded in certain settings (e.g., the 19-1 setting), the comparisons with existing methods may be skewed. They are not designed for incremental scenarios.

[1] Steering Prototypes With Prompt-Tuning for Rehearsal-Free Continual Learning.

[2] Contrastive Learning of Multivariate Gaussian Distributions of Incremental Classes for Continual Learning.

[3] Class-Incremental Mixture of Gaussians for Deep Continual Learning.

[4] Saving 100x Storage: Prototype Replay for Reconstructing Training Sample Distribution in Class-Incremental Semantic Segmentation.

**Questions:**

Please address my concerns in the 'Weakness' section.

---

### Official Review · Reviewer_ytH7 · 2024-11-08

**Soundness:** 3
**Presentation:** 2
**Contribution:** 2
**Rating:** 6
**Confidence:** 3

**Summary:**

This paper mainly focuses on the incremental object detection task. To effectively describe complex detection feature spaces, the authors propose to learn class prototypes based on Gaussian Mixture Distribution, which may better model the distribution relationships of different classes, thus facilitating the knowledge distillation procedure and incremental learning process. Furthermore, a dimension scaling progressive learning strategy and three additional loss functions are introduced to enhance the training stability and adaptability of the proposed prototype modeling method.

**Strengths:**

1. The motivation of this paper is clearly stated, and the proposed solution of modeling Gaussian Mixture Distribution based prototypes sounds reasonable.
2. The experiments under different settings are conducted and the ablation studies are comprehensive, which indicate the effectiveness of the proposed method and show the effects of introduced DSPL and DAPO strategies.
3. The implementation details, including training/optimization details and hyper-parameter settings, are provided, which can increase the reproducibility of the proposed method.

**Weaknesses:**

1. The authors claim that their method learns higher-dimension prototypes. However, I cannot understand the definition of **"high-dimension"** here, does it mean that the representation of the ROI proposal feature vector $x$, where the prototypes are built on, is of high dimensionality? But as stated in the 306th-310th rows, the introduced dimension scaling progressive learning strategy transforms the original feature vector $x$ into a **lower dimensional** vector $z$, and calculates the prototypes based on $z$. This operation seems to conflict with the concept of **"high-dimension"**, could the authors make more discussions about this?

2. In Eq. (3), the details of how to assign the ground-truth label $y_{i,c}$ for each proposal feature vector should be clarified, can we find the details of such label assignment operation in previous works?

3. From Table 5, it can be seen that the performance of the proposed method is quite sensitive to the number of Gaussian mixture components, which may increase the difficulty of tuning this hyper-parameter for different application scenarios.

4. In Section 3.2, there are some notations are confused. For example:
     1) In the 268th row, both the dimensionality of proposal feature $x_i$ and the size of input feature set are denoted as $N$.
     2) In the 276th to 277th rows, the component number of Gaussian mixture distribution is denoted as $K$, but it is also indicated by $M$ in Eq. (9) and (10).
     3) The notations of the proposal feature vector $x_i$ are bold in Eq. (4) and Eq. (7), but not in Eq. (2), (3) and (5). They should be consistent.

**Questions:**

1. For the confusion matrix in Figure 4, I am confused about the values calculated for the proposed GMDP method. Since the class prototypes of GMDP are Gaussian mixture distributions, I am not sure how to calculate the Jensen-Shannon (JS) divergence between them. Based on my knowledge, it can only get an upper bound value of this metric for a pair of Gaussian mixture distributions, so could the authors provide more details about such calculation process?

2. In the appendix A.2, I have noticed that the proposed method can also be integrated into a CL-DETR detector. Since this query-based method does not generate the proposal feature vectors, I am not very clear on how to produce the Gaussian Mixture Distribution prototypes for it. Could the authors provide more details for such integrating process?

---

> ### Comment · Reviewer_ytH7 · 2024-11-27
>
> Afther checking the resposes for my questions, I believe that most of my concerns have been resolved. Considering the originality and quality of this article, I decide to maintain my rating as “6: marginally above the acceptance threshold”.

---

### Meta-Review · Area_Chair_WAut · 2024-12-18

**Metareview:**

a) This paper presents a new method for incremental object detection (IOD) and aims to address knowledge shift using prototype learning. The authors propose a prototype based on a Gaussian Mixture Distribution Prototype (GMDP) to model the more complex feature distribution of the detection task and a Dynamic Adaptive Prototype Optimization (DAPO) strategy to improve the plasticity and stability of class prototypes. These contributions show improved results on several datasets.

b) The paper is well written with a clear motivation. The proposed solution of modelling prototypes with a Gaussian Mixture Distribution makes sense. Experiments and ablation studies are comprehensive.

c) This method is not specifically tailored for incremental object detection as it can be used for any incremental recognition method. In addition, the use of GMM is not novel although their use is different than in previous paper.

d) While the fact that the proposed method is not tailored for object detection is a negative point, in general the paper seems of great relevance with comprehensive experiments and ablations and strong empirical results. Thus, I consider that the paper should be accepted as poster.

**Additional Comments On Reviewer Discussion:**

Rev. ytH7 provided a good review, pointing out some important weaknesses, like the confusion between high dimensional representation for the prototypes vs. high number of prototypes with a mixture of Gaussians. Authors answered to the points and rev. decided to maintain their positive score.

Rev. A3Cy is the most critical for this paper. Their main points are 1) GMM are already used in previous work 2) The proposed approach is not tailored to detection 3) Not fair evaluation. Authors provided a good rebuttal, but rev. did not acknowledge the answers. In my opinion, only point 2 is not solved. However, I consider it as a minor problem. Finally, after the end of the discussion he changed their score form 3 to 5.

Rev. sEU9 provided a short review, pointing out that the paper tackles an important problem, is easy to follow and has good experimental evaluation. They provided some minor weaknesses. Authors answered well to all points, but rev. did not acknowledge the answers.

Rev. vCwD provided an initial score of 6, with positive points and only minor weaknesses. Authors answered to all questions and rev. raised their score to 8.

Overall, the paper is well written and results are comprehensive and quite positive. There are still some points that are not fully clear (e.g. how is the method tailored to detection), but overall most reviewers agreed that the paper deserves publication.

---

### Decision · Program_Chairs · 2025-01-22

Accept (Poster)